# TOWARDS A SELF-MADE MODEL: ZERO-SHOT SELF-SUPERVISED PURIFICATION FOR ADVERSARIAL ATTACKS

## ABSTRACT

Adversarial purification is an adversarial defense method without robustness training for the classifier and regardless of the form of attacks, aiming to remove the adversarial perturbations on the attacked images. Such methods can defend against various unseen threats without modifying the classifier in contrast to empirical defenses. However, previous purification methods require careful training of a strong generative model or incorporating additional knowledge when training a classifier to be comparable to adversarial training. Retraining promising generative models or classifiers on large-scale datasets (e.g., ImageNet) is extremely challenging and computation-consuming. In this work, following the natural image manifold hypothesis, we propose a zero-shot self-supervised method for adversarial purification named *ZeroPur*: For an adversarial example that lies beyond the natural image manifold, its corrupted embedding vector is first restored so that it is moved close to the natural image manifold. The embedding is then fine-tuned on finer intermediate-level discrepancies to project it back within the manifold. The whole purification process is done from coarse to fine, which does not rely on any generative model and does not require retraining the classifier to incorporate additional knowledge. Extensive experiments on three datasets including CIFAR-10, CIFAR-100, and ImageNet with various classifier architectures including ResNet and WideResNet, demonstrate that our method achieves state-of-the-art robust performance. Code released[1].

## 1 INTRODUCTION

Recent studies show that adding carefully crafted imperceptible perturbations to natural examples can easily fool deep neural networks (DNNs) to make wrong decisions Goodfellow et al. (2014); Szegedy et al. (2013). The potential vulnerability behind their remarkable performance raises a significant challenge to security-critical applications. Thus, exploring efficient adversarial defense strategies is necessary for real-world applications in DNNs.

One adversarial defense strategy that is widely considered to be efficient is *adversarial training* Jia et al. (2022); Madry et al. (2017); Zhang et al. (2019), which incorporates adversarial examples into the model training, causing the model to empirically adapt to adversarial perturbations. However, such approaches usually require huge computational resources Shafahi et al. (2019); Wu et al. (2022) and suffer from substantial performance degradation Dai et al. (2022); Kang et al. (2019); Laidlaw et al. (2020) in the presence of unseen attacks that are not involved in training.

Another adversarial defense strategy is *adversarial purification* Nie et al. (2022); Shi et al. (2021); Yoon et al. (2021), which purifies adversarial examples by removing adversarial perturbations. Unlike adversarial training, adversarial purification does not require additional adversarial examples and effectively defends against unseen attacks. The purification techniques at this stage can be roughly considered in two categories: the former uses generative models Goodfellow et al. (2014); Song & Ermon (2019); Song et al. (2020) to remove adversarial perturbations on images Nie et al. (2022); Shi et al. (2021); Wang et al. (2022a). Such methods are supported by generative models to achieve

---

[1]Code available at https://github.com/

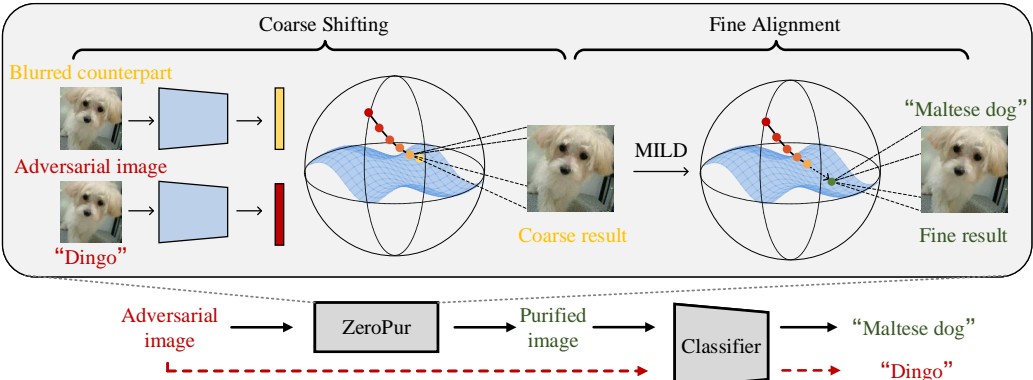

Figure 1: An illustration of ZeroPur. Given an adversarial example, we use its blurred counterpart as a reference for coarse shifting, and then fine-tune the coarse result to obtain the fine alignment result.

global image modeling and better performance than adversarial training. Yet it is often too expensive and difficult to train a promising generative model. Another purification technique uses specific lightweight pre-processing means instead of generative models Dziugaite et al. (2016); Liao et al. (2018), and thus enabling fast purification. However, the necessity to achieve comparable performance with adversarial training requires additional knowledge on the classifier, such as the auxiliary loss Mao et al. (2021); Shi et al. (2021).

In this work, following the natural image manifold hypothesis, we consider adversarial purification to move adversarial examples that are beyond the natural image manifold back onto the manifold, and propose a new adversarial purification method named ZeroPur. As illustrated in Figure 1, we move adversarial examples towards spots of the target natural images on the manifold by repeatedly pulling the distance between adversarial examples and their various blurred counterparts close in the embedding space. The movement procedure is limited by the low-quality embedding of their blurred counterparts, and cannot be precisely returned back to the original spot of target natural images on the manifold, providing only a reasonable direction. Thus, we maximize the intermediate-level discrepancies between the previous results and corresponding adversarial examples without sacrificing the direction, allowing previous results that lie near the manifold to continue moving toward it.

The main contributions of the current work are as follows:

1. We analyze the relationship between adversarial attack and adversarial purification based on the natural image manifold hypothesis, and show that a simple blurring operator can bring adversarial examples far from the natural image manifold back close.

2. We propose a zero-shot self-supervised approach for adversarial purification named ZeroPur, including two stages from coarse to fine and not depending on additional purification models.

3. The proposed approach can efficiently purify adversarial examples even does not require the classifier to learn additional knowledge. Meanwhile, the proposed approach shows superior performance if sufficient additional knowledge is available (e.g., strong data augmentation).

4. Extensive experiments demonstrate that the proposed approach outperforms current lightweight purification approaches on various datasets and has competitive performance with state-of-the-art approaches relying on generative models.

## 2 REVIEW OF LITERATURE

**Adversarial training** Adversarial training Jia et al. (2022); Luo et al.; Madry et al. (2017); Wu et al. (2020); Zhang et al. (2019) has been shown to be an effective way to improve robustness, by incorporating adversarial examples into the training data and reformulating the optimization objective from a minimization problem to a minimax problem. However, the computational cost of adversarial training is huge, caused by the fact that crafting adversarial examples requires backpropagation multiple times. In contrast to the better performance of adversarial training on seen attacks, it suffers

from substantial performance degradation in the presence of unseen attacks that are not involved in training Dai et al. (2022); Kang et al. (2019); Laidlaw et al. (2020).

**Adversarial purification** By well-designed preprocessing techniques, adversarial examples can be projected back near the natural image fold. This type of approach is called adversarial purification. Samangouei et al. (2018) propose defense-GAN, a generator that models the distribution of unperturbed images. Song et al. (2017) assume that adversarial examples are mainly low in the low probability density region of the training distribution, and design PixelDefend to approximate this distribution using the PixelCNN Van Den Oord et al. (2016). Recently, works using score-based models Yoon et al. (2021) and diffusion models Nie et al. (2022); Wang et al. (2022a) as purification models have been proposed and demonstrated to yield much better performance. These approaches first destroy adversarial perturbations with known Gaussian noise and then remove the Gaussian noise with Langevin sampling or stochastic differential equation (SDE) to achieve clean examples. Contrary to the above approaches utilizing generative models, Shi et al. (2021) propose a lightweight purification approach SOAP, which uses self-supervised loss to purification. SOAP no longer relies on generative models and can run quickly, but it requires classifiers to use the corresponding auxiliary loss in the training stage.

**Intermediate-level discrepancies** The intermediate-level discrepancies of neural networks reflect the analytical procedure of their decision, and much work on adversarial attacks and defenses has been proposed based on such discrepancies. Recently, intermediate-level discrepancies have been shown to improve the transferability of adversarial attacks Gao et al. (2021); Huang et al. (2019); Wang et al. (2021); Yan et al. (2022). Instead of distorting the output layer, such feature-level attacks maximize the distortion of intermediate-level discrepancies and achieve higher transferability. Similarly, it can be applied to adversarial defense techniques Bai et al. (2021a); Wang et al. (2022b); Zhou et al. (2021). This work discusses adversarial purification based on intermediate-level discrepancies.

## 3 PROPOSED ZERO-SHOT LEARNING FOR ADVERSARIAL PURIFICATION

### 3.1 ADVERSARIAL ATTACKS AND PURIFICATION IN NATURAL IMAGE MANIFOLD HYPOTHESIS

In the natural image manifold hypotheis Ho et al. (2022), all natural images lie on a specific manifold called the natural image manifold. The learning process for DNNs can be considered as modeling this manifold, and its embedding space is an approximation of the natural image manifold.

Considering an embedding function $f$ and a decision function $g$, the natural image $\mathbf{x}$ is embedded by $f(x) \in \mathbb{R}^d$ to the manifold $\mathcal{M}$, where $\mathbf{y} \in \mathbb{R}$ is its label. Adversarial attacks aim to move $\mathbf{x}$ beyond the manifold by optimizing the following objective with classification loss function $\ell$:

$$\max_{\|\delta^*\| \leq \epsilon} \ell(g \circ f(\mathbf{x} + \delta^*), \mathbf{y}),\tag{1}$$

where $\epsilon$ is the budget of adversarial perturbation. In most contexts, $\delta^*$ is approximated by the local worst-case $\delta$. For example, The Projected Gradient Descent (PGD) Madry et al. (2017) updates adversarial perturbations in each step with the following equation:

$$\delta^{t+1} = \Pi_\epsilon(\delta^t + \alpha \cdot \mathrm{sgn}(\nabla_{\delta^t} \ell(g \circ f(\mathbf{x} + \delta^t), \mathbf{y}))), \qquad t \in [0, \tau - 1],\tag{2}$$

where $\Pi_\epsilon$ is the projection operator which projects the perturbation $\delta^t$ back to the $\epsilon$-ball to measure perceptibility and $\alpha, \tau$ denote the step size and the number of iterations of attacks. Finally $\delta^t$ is then an approximation of $\delta^*$ and notated as $\delta$.

Intuitively, optimizing Equation 1 results in a large shift in the embedding for $f$, which is equivalent to the natural image $\mathbf{x}$ deviating from the manifold. We can formulate the procedure:

$$\min_{\delta^*} \|\delta^*\| \quad \text{s.t.} \quad \|f(\mathbf{x}) - f(\mathbf{x} + \delta^*)\| \geq \gamma, \ g \circ f(\mathbf{x}) \neq g \circ f(\mathbf{x} + \delta^*).\tag{3}$$

Adversarial purification can be naturally considered in the reverse process of adversarial attacks. Thus the goal of adversarial purification is to move adversarial examples that deviate from the manifold to their initial spot on the manifold. The objective of adversarial purification can be written:

$$\min_{\delta_{\mathrm{pfy}}} \|f(\mathbf{x}_{\mathrm{adv}} + \delta_{\mathrm{pfy}}) - f(\mathbf{x})\| \quad \text{s.t.} \quad \|\delta_{\mathrm{pfy}}\| \leq \epsilon_{\mathrm{pfy}},\tag{4}$$

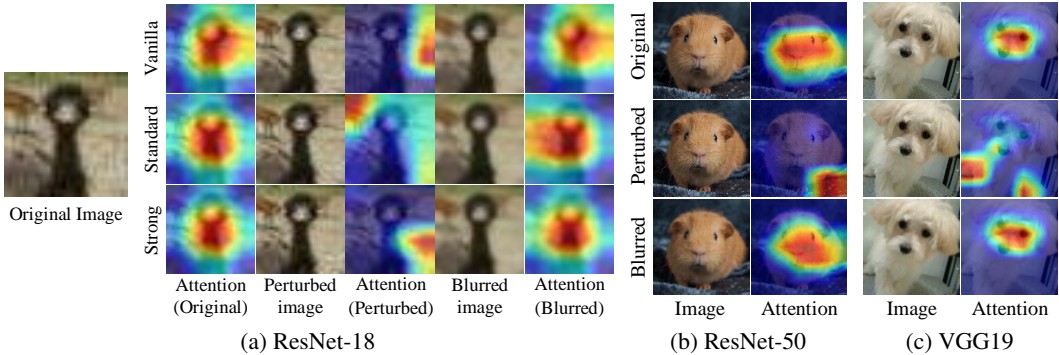

Original Image

(a) ResNet-18     (b) ResNet-50     (c) VGG19

Figure 2: An illustration of restoration on CIFAR-10 and ImageNet. Left: PGD-20 on CIFRA-10. Middle and right: PGD-20 on ImageNet. Adversarial examples distort the capacity of all networks to capture features, but this perturbation was restored after being blurred.

Table 1: Responses of various models on CIFAR-10 for different types of examples.

| Examples | Cross-Entropy Loss | | | Accuracy (%) | | |
|---|---|---|---|---|---|---|
| | Vanilla | Standard | Strong | Vanilla | Standard | Strong |
| Natural | 0.588 | 0.287 | 0.278 | 83.81 | 92.70 | 90.89 |
| Perturbed | 36.018 | 54.596 | 35.049 | 0.00 | 0.00 | 0.00 |
| Blurred | 7.475 | **3.758** | **1.037** | 11.39 | 27.70 | 68.61 |
| Coarse | 5.837 | 4.989 | 1.076 | **53.73** | **56.21** | **77.64** |

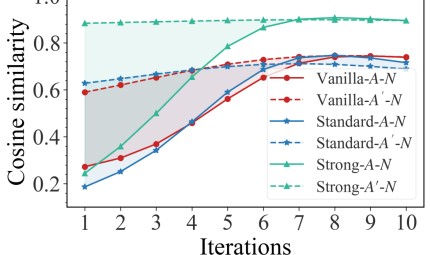

Figure 3: Details of the coarse purification on CIFAR-10.

where $\mathbf{x}_{\mathrm{adv}} = \mathbf{x} + \delta^*$, and $\mathbf{x}_{\mathrm{adv}} + \delta_{\mathrm{pfy}}$ is the idealized purification result. Similar to adversarial attacks, Equation 4 can be minimized indirectly by optimizing the following equation:

$$\min_{\delta_{\mathrm{pfy}}} \ell(g \circ f(\mathbf{x}_{\mathrm{adv}} + \delta_{\mathrm{pfy}}), \mathbf{y}) \quad \text{s.t.} \quad \|\delta_{\mathrm{pfy}}\| \le \epsilon_{\mathrm{pfy}} , \tag{5}$$

where $\delta_{\mathrm{pfy}}$ and $\epsilon_{\mathrm{pfy}}$ are defined to correspond to $\delta$ and $\epsilon$ in Equation 1 to offset perturbations.

However, the natural example $\mathbf{x}$ and its label $\mathbf{y}$ are invisible, and the only one we can use is the adversarial example $\mathbf{x}_{\mathrm{adv}}$. We are thus required to devise a purification loss function $\ell_{\mathrm{pfy}}$ with $\mathbf{x}_{\mathrm{adv}}$ taken for input, optimizing the following equation:

$$\min_{\delta_{\mathrm{pfy}}} \ell_{\mathrm{pfy}}(f(\mathbf{x}_{\mathrm{adv}} + \delta_{\mathrm{pfy}}); \Theta) \quad \text{s.t.} \quad \|\delta_{\mathrm{pfy}}\| \le \epsilon_{\mathrm{pfy}} . \tag{6}$$

where $\Theta$ are the additional parameters introduced in the design $\ell_{\mathrm{pfy}}$. Purification methods based on generative models Ho et al. (2022); Nie et al. (2022); Samangouei et al. (2018); Yoon et al. (2021) usually train a purification model $\mathcal{G}$ to minimize the global $\ell_{\mathrm{pfy}}$. Other lightweight purification methods Mao et al. (2021); Shi et al. (2021) that do not use generative models usually design a suitable $\ell_{\mathrm{pfy}}$ to indirectly minimize Equation 4, and they usually need to retrain the classifier to introduce additional knowledge. These methods all introduce learnable parameters $\Theta$ in $\ell_{\mathrm{pfy}}$. We now discuss how to design $\ell_{\mathrm{pfy}}$ without $\Theta$.

Considering that image transformation can also shift the position of embeddings $f(\mathrm{x})$ in manifold $\mathcal{M}$, we investigate the response of various classifiers to the transformed adversarial examples. We trained three ResNet-18 He et al. (2016) on CIFAR-10 with different levels of data augmentation to verify the prevalence of the phenomenon. Specifically, 'Vanilla' denotes no data augmentation, 'Standard' denotes basic data augmentation (random resized crop, random horizontal flip), and 'Strong' denotes strong data augmentation used in contrastive learning (color jitter, grayscale, gaussian blur, solarization, equalization). See Appendix A.1 for details.

As illustrated in Figure 2(a), all three networks do not capture features correctly on PGD attack Madry et al. (2017). However, after blurring the image using a median filter with a window size of $3 \times 3$, any

of them recaptures the same features as the original image. And the stronger the data augmentation, the greater the similarity of the recaptured features. The same restoration is shown in Figure 2(b) and (c) on ImageNet, where the pre-trained ResNet-50 and VGG19 (Timm Wightman (2019) version) captures features with high overlap with the original image on the blurred adversarial examples (gaussian blur with $\sigma = 1.2$). Note that they benefit from a well-designed data augmentation strategy, thus the restoration is effective. (e.g., AugMix Hendrycks et al. (2019)). Table 1 also shows the loss and robustness accuracy of natural examples, perturbed (adversarial) examples, blurring adversarial examples, and coarse purification results by our method on three classifiers. The loss w.r.t adversarial examples is significantly larger than their blurring counterparts. And the accuracy is also slightly improved, although there is still a gap for natural examples.

## 3.2 COARSE SHIFTING

Motivated by this heuristic phenomenon above that the blurring operator restores the attention of classifiers, and the loss added by worst-case adversarial examples and destroyed decisions will be fixed, we can conclude that adversarial examples that deviated from the natural image manifold are coming back closer to the manifold. Therefore, we can try to move the adversarial example to the vicinity of the natural image fold by pulling the distance between them and their blurred counterpart in the embedding space. Moreover, to avoid a single example being too blurred to be recognized by the classifier, we suggest that the blurring of a single time for adversarial examples should be attenuated, and the distance should be iteratively closed (See the appendix A.5 for a detailed discussion). The distance of feature embeddings is defined by the Cosine Similarity:

$$d(\mathbf{z}_{\mathrm{adv}}, \mathbf{z}'_{\mathrm{adv}}) = \frac{\mathbf{z}_{\mathrm{adv}} \cdot \mathbf{z}'_{\mathrm{adv}}}{\|\mathbf{z}_{\mathrm{adv}}\| \|\mathbf{z}'_{\mathrm{adv}}\|} \ , \tag{7}$$

where $\mathbf{z}'_{\mathrm{adv}}$ is the embedding of blurred adversarial example.

The workflow of coarse purification is shown in Algorithm 1. Note that $\alpha_c$ and $\epsilon_c$ are hyper-parameters of the algorithm, which in practice are empirically set to $\alpha_c = \alpha, \epsilon_c = 1.25\epsilon$. The main results of the algorithm are reported in Table 1. The accuracy is significantly improved after purification. The increase is $42.34\%$ on 'Vanilla' and $28.51\%$ on 'Standard' compared to the blurring operator. The detailed procedure of the coarse purification is shown in Fig. 3, solid lines denote the cosine similarity of features embedding between all adversarial examples and natural examples on CIFAR-10. And dashed lines denote cosine similarity between blurring adversarial examples and natural examples. Each color denotes ResNet-18 of various

---

**Algorithm 1** Coarse Shifting

**Input:** Adversarial example $\mathbf{x}_{\mathrm{adv}}$, number of iterations $K_c$, step size $\alpha_c$, a classifier $f$, purification bound $\epsilon_c$, and the blurring operator $g$.

**Output:** Coarse purification example $\mathbf{x}'$

1: Random start $\mathbf{x}_{\mathrm{adv}} \leftarrow \mathbf{x}_{\mathrm{adv}} + \varepsilon$
2: **for** $k = 1, 2, ..., K_c$ **do**
3: $\quad \mathbf{x}'_{\mathrm{adv}} \leftarrow g(\mathbf{x}_{\mathrm{adv}})$
4: $\quad \mathbf{z}_{\mathrm{adv}} \leftarrow f(\mathbf{x}_{\mathrm{adv}})$
5: $\quad \mathbf{z}'_{\mathrm{adv}} \leftarrow f(\mathbf{x}'_{\mathrm{adv}})$
6: $\quad \mathbf{x}_{\mathrm{adv}} \leftarrow \mathbf{x}_{\mathrm{adv}} + \alpha_c \cdot \mathrm{sgn}(\nabla d(\mathbf{z}_{\mathrm{adv}}, \mathbf{z}'_{\mathrm{adv}}))$
7: $\quad \mathbf{x}_{\mathrm{adv}} \leftarrow \Pi_{\mathbf{x}_{\mathrm{adv}}, \epsilon_c}(\mathbf{x}_{\mathrm{adv}})$
8: **end for**
9: $\mathbf{x}' \leftarrow \mathbf{x}_{\mathrm{adv}}$

---

data augmentation strategies. The steady increase of solid lines implies that each blurring is well guided by the purification, and the decrease of areas between solid and dashed lines also implies that the discrepancy between adversarial examples and their blurring counterparts in the embedding space is getting smaller.

## 3.3 FINE ALIGNMENT

The result of the coarse shifting is already too promising for the classifier with a strong data augmentation strategy. Once there is no aggressive data augmentation to support the classifier training, the coarse shifting is limited by the low-quality embedding of the blurred image and cannot move to the exact spot in the manifold that its corresponding natural image is in. But at least the direction of shifting is reasonable. It is demonstrated by the red and blue lines that eventually converge to a straight line in Figure 3.

Our goal turns to break the limitation of low-quality embedding of the blurred image without the support of aggressive data augmentation. Breaking the limitation and allowing the example to shift

independently is similar to Intermediate Level Attack (ILA) Huang et al. (2019). Specifically, given a function $f_l$ denoted as feature maps at layer $l$ of the classifier, we define the following objective:

$$\max_{\mathbf{x}''} \ \mathcal{L}_l(\mathbf{x}_{\mathrm{adv}}, \mathbf{x}', \mathbf{x}'') = -\Delta \mathbf{u}_l'' \cdot \Delta \mathbf{u}_l' \,, \tag{8}$$

where $\mathbf{x}''$ is the new purification result after fine-tuning, $\Delta \mathbf{u}_l''$ and $\Delta \mathbf{u}_l'$ are two vectors of flattened feature maps defined as follows:

$$\Delta \mathbf{u}_l'' = f_l(\mathbf{x}'') - f_l(\mathbf{x}_{\mathrm{adv}}) \,, \tag{9}$$

$$\Delta \mathbf{u}_l' = f_l(\mathbf{x}') - f_l(\mathbf{x}_{\mathrm{adv}}) \,. \tag{10}$$

The resulting $\mathbf{x}''$ is initialized by $\mathbf{x}_{\mathrm{adv}}$. Maximizing Equation 8 is equivalent to maximizing the projection of $\mathbf{u}_l''$ on $\mathbf{u}_l'$ since $\|\mathbf{u}_l'\|$ is a constant. The increase of projection implies that $x''$ is not restricted by the blurring example to continue moving along the direction of the coarse result, which makes $x''$ hopefully move independently to the natural manifold. It allows us to refine the purification result by making each pixel change significant at a constrained purification budget $\delta_{\mathrm{pfy}}$. The fine-tuning process is called fine alignment.

We empirically found that using feature maps deep in the classifier can significantly improve alignment results, and using multiple layers will yield better results than single layers. We therefore designed Multiple Intermediate-Level Discrepancies (MILD). Let $L = \{l_1, l_2, ..., l_m\}$ denote the set of a $m$ layers model $f$, we carefully selected $S \subseteq L$ as a candidate for computing Equation 8. The MILD can be rewritten as:

$$\max_{\mathbf{x}''} \ \mathrm{MILD}(\mathbf{x}_{\mathrm{adv}}, \mathbf{x}', \mathbf{x}'') = \frac{1}{\|S\|} \sum_{l \in S} \mathcal{L}_l(\mathbf{x}_{\mathrm{adv}}, \mathbf{x}', \mathbf{x}'') \quad \text{s.t.} \quad \|\mathbf{x}'' - \mathbf{x}_{\mathrm{adv}}\| \le \epsilon_{\mathrm{pfy}} \,, \tag{11}$$

where $\|S\|$ denotes the number of elements of the set $S$.

Algorithm 2 describes details of the fine alignment. In line 2 we estimate the step size $\alpha_f$ of each iteration with iteration number $K_f$ and purification budget $\epsilon_f$, not wasting any pixels altered by each alignment. In practice, we consider the last three layers of the classifier into the candidate set $S$. The intuitive understanding of the approach is: Each iteration allows the purified example to move autonomously in the direction of the previous coarse shifting, but the process is unaffected by other low-quality embeddings and thus can reach the original natural image spot successfully. The results of fine alignment are reported in Section 4. Not surprisingly, the direction of fine alignment is strongly correlated with coarse shifting. Algorithm 2 backfires if coarse shifting does not give an approximately correct direction as a reference for fine alignment.

---

**Algorithm 2** Fine Alignment

**Input:** Adversarial example $\mathbf{x}_{\mathrm{adv}}$, coarse shifting result $\mathbf{x}'$, number of iterations $K_f$, a classifier $f$, purification bound $\epsilon_f$, and candidate layers $S$.

**Output:** Final purification results $\mathbf{x}''$

1: Initialize $\mathbf{x}'' \leftarrow \mathbf{x}_{\mathrm{adv}}$
2: Set step size $\alpha_f \leftarrow \epsilon_f / K_f$
3: **for** $k = 1, 2, ..., K_f$ **do**
4:      **for** $l \in S$ **do**
5:          $\Delta \mathbf{u}_l' \leftarrow f_l(\mathbf{x}') - f_l(\mathbf{x}_{\mathrm{adv}})$
6:          $\Delta \mathbf{u}_l'' \leftarrow f_l(\mathbf{x}'') - f_l(\mathbf{x}_{\mathrm{adv}})$
7:          $\mathcal{L} \leftarrow \mathcal{L} + (-\Delta \mathbf{u}_l'' \cdot \Delta \mathbf{u}_l')$
8:      **end for**
9:      $\mathbf{x}'' \leftarrow \mathbf{x}'' - \alpha_f \cdot \mathrm{sgn}(\nabla_{\mathbf{x}''} \frac{1}{\|S\|} \mathcal{L})$
10:      $\mathbf{x}'' \leftarrow \Pi_{\mathbf{x}'', \epsilon_f}(\mathbf{x}'')$
11: **end for**

---

## 4 EXPERIMENTS

### 4.1 EXPERIMENTAL SETTINGS

**Datasets and base classifier.** Three benchmark datasets CIFAR-10 Krizhevsky et al. (2009), CIFAR-100 and ImageNet Deng et al. (2009) were considered to evaluate the robustness. We compare with the state-of-the-art adversarial training methods reported in standard benchmark RobustBench Croce et al. (2021) and other adversarial purification methods on such three datasets. We consider the based model ResNet-18 He et al. (2016) and WideResNet-28-10 Zagoruyko & Komodakis (2016) on CIFAR-10 and CIFAR-100, and ResNet-50 on ImageNet. As described in Section 3.2, we consider various data augmentation strategies to train based models on CIFAR-10 and CIFAR-100 (See Appendix A.1 for

details), demonstrating the positive of additional data augmentation on our approach. In practice, we use a median filter with $3 \times 3$ window size as the blurring operator on the 'Vanilla' classifier, and Gaussian blur with $\sigma = 1.2$ on the 'Standard' and 'Strong' classifiers. For simplicity, we use the notation 'V', 'B', and 'S' to denote the results on 'Vanilla', 'Standard', and 'Strong', while 'C' and 'F' denote the results on coarse shifting and fine alignment. For example, 'ZeroPur-B-C-F' denotes the results for ZeroPur with fine alignment on 'Standard'.

**Adversarial attacks and evaluation metrics.** We evaluate our method with standard attacks and strong adaptive attacks. For standard attacks where the defense strategy is unknown to the adversary, we use PGD attack and AutoAttack Croce & Hein (2020) with adversarial training methods and other adversarial purification methods. For strong adaptive attacks, the adversary knows the defense strategy for the model. We use Defense Aware (DA) Attack Mao et al. (2021) and BPDA+EOT Athalye et al. (2018); Hill et al. (2021) to evaluate our method, where BPDA+EOT is the strongest attack for purification methods so far.

## 4.2 COMPARISON WITH THE STATE-OF-THE-ART

**CIFAR-10 & CIFAR-100** Table 2 and Table 3 reports the robust performance against $\ell_\infty$ threat model ($\epsilon = 8/255$) and $\ell_2$ threat model ($\epsilon = 0.5$) with PGD-20 and AutoAttack on CIFAR-10, as well as Table 4 on CIFAR-100. 'Training Required' denotes that the method requires retraining the classifier. It can be seen that our method yields better robust performance than previous state-of-the-art methods even without robustness training in $\ell_\infty$ threat model. In $\ell_2$ threat model, our method is also comparable to state-of-the-art methods. Meanwhile, classifiers with data augmentation ('Strong') obtain greater robustness. For a fair comparison, we regard the 'Strong' as Training Required.

Table 2: Robust accuracy (%) against PGD-20 and AutoAttack $\ell_\infty(\epsilon = 8/255)$ on CIFAR-10, obtained by different classifier architectures. The first part corresponds to adversarial training methods and the second part corresponds to adversarial purification methods.

| Training Required | ResNet-18 | | | WideResNet-28-10 | | |
| :---: | :--- | :---: | :---: | :--- | :---: | :---: |
| | Method | PGD-20 | AutoAttack | Method | PGD-20 | AutoAttack |
| ✓ | (Gowal et al., 2021) | 61.31 | 59.12 | (Gowal et al., 2021) | 66.09 | 63.99 |
| ✓ | (Sehwag et al., 2021) | 59.00 | 56.19 | (Pang et al., 2022) | 64.92 | 61.47 |
| ✓ | (Rade et al., 2021) | 61.71 | 58.17 | (Gowal et al., 2020) | 66.05 | 63.27 |
| ✓ | (Addepalli et al., 2022b) | 56.71 | 52.90 | (Rade et al., 2021) | 66.04 | 63.36 |
| ✓ | (Shi et al., 2021) | 60.65 | 66.62 | (Shi et al., 2021) | 65.43 | 68.56 |
| ✓ | (Mao et al., 2021) | 54.59 | 58.20 | (Mao et al., 2021) | 64.64 | 67.79 |
| ✗ | ZeroPur-V-C | 53.73 | 55.58 | ZeroPur-V-C | 57.41 | 58.21 |
| ✗ | ZeroPur-V-C-F | 69.52 | 68.59 | ZeroPur-V-C-F | 67.82 | 67.20 |
| ✗ | ZeroPur-B-C | 56.21 | 58.66 | ZeroPur-B-C | 57.45 | 53.29 |
| ✗ | ZeroPur-B-C-F | **69.56** | **71.76** | ZeroPur-B-C-F | **70.66** | **69.39** |
| ✓ | ZeroPur-S-C | 77.64 | 79.29 | ZeroPur-S-C | 76.77 | 78.82 |
| ✓ | ZeroPur-S-C-F | 85.15 | 83.46 | ZeroPur-S-C-F | 83.92 | 82.31 |

**ImageNet** Table 5 shows the robust performance against $\ell_\infty$ threat model ($\epsilon = 4/255$) with PGD-200 and AutoAttack on ImageNet. The upper of the table shows adversarial training methods, which require classifier robustness training. In the middle, adversarial purification methods including DISCO, DiffPure and GDMP all rely on purification models, while Reverse Attack in the bottom requires classifier robust training. However, Our method achieves similar robustness to them even without relying on any purification models or retraining. We provide visual examples in Figure 4 to show how our method purifies the adversarial examples. Note that Reverse Attack is used as a post-processing technique for boosting adversarial training methods, thus we report its best performance on ImageNet.

## 4.3 DEFEND AGAINST STRONG ADAPTIVE ATTACKS

Assuming that the adversary is aware of the specific defense method for adversarial purification, strong adaptive attacks can be conducted. Thus we evaluate the robustness of ZeroPur on adaptive attacks including BPDA+EOT Athalye et al. (2018); Hill et al. (2021) and DA Attack Mao et al. (2021) on

Table 3: Robust accuracy (%) against AutoAttack $\ell_2(\epsilon = 0.5)$ on CIFAR-10. The order of method types is consistent with Table 2. (Accuracy not reported in respective papers is replaced by '-'.)

| Method | Tra. | Arch. | Robust (%) PGD-20 | AA |
|---|---|---|---|---|
| (Rade et al., 2021) | ✓ | ResNet-18 | 77.48 | 76.30 |
| (Rebuffi et al., 2021) | ✓ | ResNet-18 | 78.08 | 76.06 |
| (Sehwag et al., 2021) | ✓ | ResNet-18 | 76.11 | 74.59 |
| (Sehwag et al., 2021) | ✓ | WRN-34-10 | 79.03 | 77.41 |
| (Augustin et al., 2020) | ✓ | WRN-34-10 | **81.35** | 78.98 |
| (Wu et al., 2020) | ✓ | WRN-34-10 | 75.33 | 73.75 |
| (Sun et al., 2019) | ✓ | WRN-28-10 | - | 74.33 |
| (Nie et al., 2022) | ✓ | WRN-28-10 | - | 78.58 |
| (Nie et al., 2022) | ✓ | WRN-70-16 | - | 80.60 |
| (Ho et al., 2022) | ✓ | WRN-28-10 | - | **88.47** |
| ZeroPur-V-C-F | ✗ | ResNet-18 | 74.89 | 74.23 |
| ZeroPur-B-C-F | ✗ | ResNet-18 | 79.21 | 80.76 |
| ZeroPur-S-C-F | ✓ | ResNet-18 | 90.85 | 89.07 |
| ZeroPur-V-C-F | ✗ | WRN-28-10 | 76.59 | 74.84 |
| ZeroPur-B-C-F | ✗ | WRN-28-10 | 77.89 | 80.37 |
| ZeroPur-S-C-F | ✓ | WRN-28-10 | 89.77 | 85.00 |

Table 4: Robust accuracy (%) against PGD-20 and AutoAttack $\ell_\infty(\epsilon = 8/255)$ on CIFAR-100, obtained by different classifier architectures. The order of method types is consistent with Table 2.

| Method | Tra. | Arch. | Robust (%) PGD-20 | AA |
|---|---|---|---|---|
| (Rade et al., 2021) | ✓ | ResNet-18 | 32.71 | 29.50 |
| (Addepalli et al., 2022a) | ✓ | ResNet-18 | 33.29 | 27.83 |
| (Addepalli et al., 2022b) | ✓ | ResNet-18 | 34.04 | 28.58 |
| (Rebuffi et al., 2021) | ✓ | WRN-28-10 | 36.11 | 33.03 |
| (Pang et al., 2022) | ✓ | WRN-28-10 | 35.48 | 31.67 |
| (Jia et al., 2022) | ✓ | WRN-34-10 | 36.45 | 32.17 |
| (Shi et al., 2021) | ✓ | ResNet-18 | 28.67 | 34.38 |
| (Mao et al., 2021) | ✓ | ResNet-18 | 23.83 | 25.45 |
| (Shi et al., 2021) | ✓ | WRN-28-10 | 37.66 | 37.57 |
| (Mao et al., 2021) | ✓ | WRN-34-10 | 31.21 | 33.16 |
| ZeroPur-V-C-F | ✗ | ResNet-18 | 36.64 | 34.40 |
| ZeroPur-B-C-F | ✗ | ResNet-18 | **37.61** | **37.75** |
| ZeroPur-S-C-F | ✓ | ResNet-18 | 55.43 | 54.05 |
| ZeroPur-V-C-F | ✗ | WRN-28-10 | 32.56 | 32.93 |
| ZeroPur-B-C-F | ✗ | WRN-28-10 | 34.80 | 35.65 |
| ZeroPur-S-C-F | ✓ | WRN-28-10 | 55.89 | 51.83 |

Table 5: Natural accuracy and robust accuracy (%) against $\ell_\infty$ threat model ($\epsilon = 4/255$) on ImageNet, obtained by ResNet-50. In our method, the blurring operator is Gaussian blur with $\sigma = 1.2$. The $\dagger$ indicates evaluation with PGD-200, otherwise, evaluation with AutoAttack. (The accuracy is directly reported from the respective paper.)

| Method | Model Required | Training Required | Accuracy (%) Natural | AutoAttack |
|---|---|---|---|---|
| (Salman et al., 2020) | ✗ | ✓ | 64.02 | 34.96 |
| (Wong et al., 2020) | ✗ | ✓ | 55.62 | 26.24 |
| (Bai et al., 2021b) | ✗ | ✓ | 67.38 | 35.51 |
| DISCO (Ho et al., 2022) | LIIF | ✗ | **71.22** | **69.52** |
| DiffPure (Nie et al., 2022) | SDE | ✗ | 67.79 | 40.93 |
| GDMP (Wang et al., 2022a) | Guided DDPM | ✗ | 70.17 | 68.78 |
| Reverse Attack$^\dagger$ (Mao et al., 2021) | ✗ | ✓ | - | 31.32 |
| ZeroPur (Ours) | ✗ | ✗ | 62.15 | 67.54 |

CIFAR-10 with WideResNet-28-10. Figure 5 reports the robustness of different purification steps on two strong adaptive attacks. The robustness of our method is stable on DA Attack, but on BPDA+EOT attack decreases as the number of purification steps increases. It could be caused by the increase in purification steps that lead to the correct estimation of the gradient by BPDA+EOT. In Table 6, we show the comparison with other purification methods on BPDA+EOT attack. Purification methods based sampling loops such as Hill et al. (2021) have naturally stronger defense on BPDA+EOT attack, and our method is slightly below it. The robustness of our method against strong adaptive attacks decreases partly because these attacks start from clean natural examples. However, our method is based on the assumption that the purified images are adversarial images (as described in the Appendix A.4). We believe that the direct use of strong adaptive attacks underestimates the robustness of ZeroPur. Following the comparison criteria of most of the literature, but, we still report all results.

## 4.4 DISCUSSION FOR BLURRING OPERATORS IN ZEROPUR

We show how the blurring operators affect the robust performance in Appendix A.3, evaluating the robustness of our method on CIFAR-10 using different blurring operators including median filters and Gaussian blurring kernels. The results show that the excessive blurring may cause the 'Vanilla'

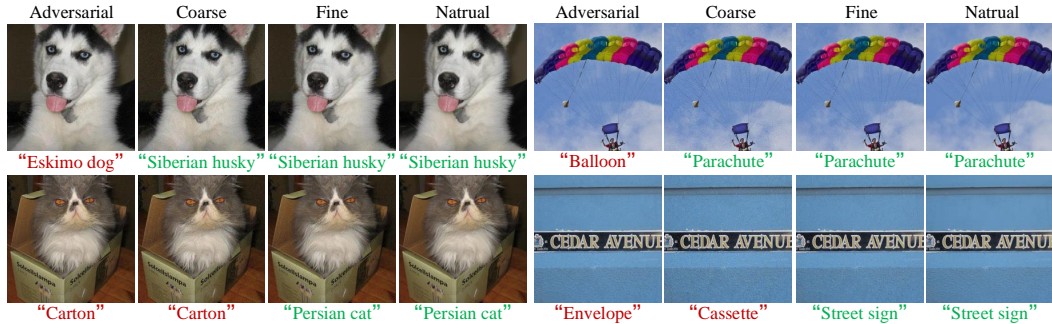

Figure 4: Visual examples of ZeroPur against $\ell_\infty$ threat model ($\epsilon = 8/255$) on ImageNet. The red label is the error prediction and the green label is the correct prediction.

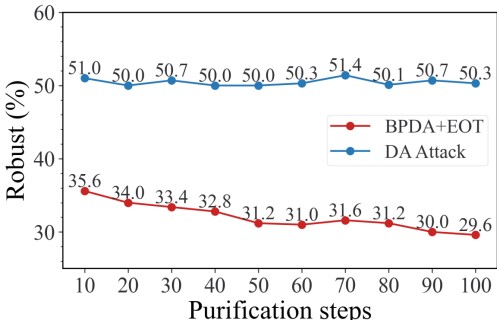

Figure 5: Impact of purification steps in our method on Robust accuracies.

Table 6: Comparison of robust accuracy (%) with other adversarial purification methods using the BPDA+EOT with $\ell_\infty(\epsilon = 8/255)$ threat model.

| Method | Purification | BPDA+EOT |
|---|---|---|
| (Song et al., 2017) | Gibbs Update | 9.00 |
| (Yang et al., 2019) | Mask+Recon. | 15.00 |
| (Hill et al., 2021) | EBM+LD | **54.90** |
| (Shi et al., 2021) | Auxiliary Loss | 31.90 |
| ZeroPur-B-C | Zero-shot | 35.60 |
| ZeroPur-B-C-F | Zero-shot | 25.00 |

classifier to recognize images incorrectly. On the contrary, aggressive blurring operators achieve better robust performance on classifiers with strong data augmentation.

ZeroPur utilizes the blurring operator to move adversarial examples back to the natural image manifold, which follows that such adversarial examples are not robust to blurring operations. However, some attacks are naturally robust to blurring operations, such as DI$^2$-FGSM Xie et al. (2019). It does not mean that these attacks are completely immune to ZeroPur, and we can replace the blurring operator with other operators that destroy adversarial examples to further strengthen the purification, i.e., TV Minimization Guo et al. (2017). The purification results of ResNet-18 on CIFAR-10 before and after the replacement are reported in Table 7, and the performance against PGD attacks rises to 77.03%, which even surpasses the optimal performance achieved by blurring.

Table 7: Robust accuracy (%) against PGD and DI$^2$-FGSM by blurring and TV Minimization. The Better performance in each attack is bolded.

| Method | PGD | | DI$^2$-FGSM | |
|---|---|---|---|---|
| | Blurring | TV | Blurring | TV |
| ZeroPur-V-C | **53.73** | 51.72 | 35.38 | **39.17** |
| ZeroPur-V-C-F | 69.52 | **69.53** | 57.86 | **61.85** |
| ZeroPur-B-C | 56.21 | **63.27** | 40.42 | **49.52** |
| ZeroPur-B-C-F | 69.56 | **77.03** | 60.48 | **69.96** |
| ZeroPur-S-C | **77.64** | 65.44 | **68.13** | 53.50 |
| ZeroPur-S-C-F | **85.15** | 72.31 | **82.34** | 66.39 |

## 5 CONCLUSIONS

We propose a zero-shot self-supervised method for adversarial purification named *ZeroPur* that does not rely on any generative model or does not require retraining the classifier to incorporate additional knowledge. Our method largely outperforms previous state-of-the-art adversarial training and adversarial purification methods and is more lightweight.

Despite the improvements, our method has two major limitations: (i) our method suffers when applied to clean natural images because the blurring operator corrupts the clean image. (ii) it is not feasible to completely eliminate the distortion on images. We leave them for our future work.

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

# A APPENDIX

## A.1 IMPLEMENTATION DETAILS OF CLASSIFIERS

Our experiments on CIFAR-10 Krizhevsky et al. (2009) and CIFAR-100 including three classifiers 'Vanilla', 'Standard', and 'Strong', were trained with three different data augmentation strategies. Table 8 shows the training settings and natural accuracy. 'Standard' that most of the adversarial training methods and adversarial purification methods are using has better natural accuracy.

| Classifier | Name | Trained On | | | | | | | Natural Accuracy (%) | |
|---|---|---|---|---|---|---|---|---|---|---|
| | | ReCrop. | ColorJ. | GrayS. | GauBlur. | Solar. | Equal. | HorFlip. | CIFAR-10 | CIFAR-100 |
| ResNet-18 | Vanilla | | | | | | | | 83.81 | 56.60 |
| | Standard | ✓ | | | | | | ✓ | **92.70** | **71.68** |
| | Strong | ✓ | ✓ | ✓ | ✓ | ✓ | ✓ | ✓ | 90.89 | 68.49 |
| WRN-28-10 | Vanilla | | | | | | | | 91.55 | 71.50 |
| | Standard | ✓ | | | | | | ✓ | **93.91** | **74.95** |
| | Strong | ✓ | ✓ | ✓ | ✓ | ✓ | ✓ | ✓ | 91.08 | 67.52 |

Table 8: Training details and Natural Accuracy (%) for base classifiers on CIFAR-10 and CIFAR-100. The enumerated data augmentation are, in order, ResizeCrop, ColorJitter, Grayscale, Solarization, Equalization, and HorizontalFlip.

All classifiers were trained by the SGD optimizer with a cosine decay learning rate schedule Loshchilov & Hutter (2016) and a linear warm-up period of 10 epochs. The weight decay is $5.0 \times 10^{-4}$ and the momentum is $0.9$. The initial learning rate is set to $0.1$. Classifiers were trained for 120 epochs on 4 Tesla V100 GPUs, where the batch size is 512 per GPU for ResNet-18 and 128 per GPU for WideResNet-28-10.

## A.2 IMPLEMENTATION DETAILS OF ADVERSARIAL ATTACKS

**PGD** Madry et al. (2017) We use PGD attacks implemented by `Foolbox` Rauber et al. (2017). We set $\epsilon = 8/255, \alpha = 2/255$ and $\epsilon = 0.5, \alpha = 0.05$ for PGD-20 $\ell_\infty$ and $\ell_2$ threat model, and the iterations is 20 on CIFAR-10 and CIFAR-100. For PGD-20 $\ell_\infty$ on ImageNet Deng et al. (2009), we set $\epsilon = 4/255$. `Foolbox` available at https://github.com/bethgelab/foolbox.

**AutoAttack** Croce & Hein (2020) We use AutoAttack to compare with the start-of-the-art methods. The robust classifier for adversarial training methods provided by RobustBench Croce et al. (2021) benchmark that available at https://robustbench.github.io. The code for adversarial purification methods is provided by their respective papers.

There are two versions of AutoAttack: (i) the STANDARD including AGPD-CE, AGPD-T, FAB-T, and Square, and (ii) the RAND version including APGD-CE and APGD-DLR. Considering that most of the adversarial purifications choose the RAND version, all the performance in this work we report is also in the RAND version. Code available at https://github.com/fra31/auto-attack.

**Strong adaptive attacks** There are two adaptive attacks evaluating our method including Defense Aware (DA) attack Mao et al. (2021) and BPDA+EOT attack Hill et al. (2021). To make a fair comparison, we use their codebase: https://github.com/cvlab-columbia/SelfSupDefense; https://github.com/point0bar1/ebm-defense with default hyperparameters for evaluation.

## A.3 DETAILED DISCUSSION FOR BLURRING OPERATIORS

We show how the blurring operators affect the robust performance in Figure 9. We evaluate the robustness performance of our method on CIFAR-10 using 5 different blurring operators including median filters with windows sizes $3 \times 3$ and $5 \times 5$ and Gaussian blurring kernels with $\sigma = 0.6, 1.2, 1.8$. We can see that if the classifier is not adapted to aggressive data augmentation, excessive blurring

of the adversarial examples should be avoided in purification, and excessive blurring may cause the 'Vanilla' classifier to fail to recognize images correctly. On the contrary, aggressive blurring operators can have better robust performance on classifiers with strong data augmentation.

| Architecture | Method | Median Filter | | | Gaussian Blur | | |
|---|---|---|---|---|---|---|---|
| | | $3 \times 3$ | $5 \times 5$ | $7 \times 7$ | $\sigma = 0.6$ | $\sigma = 1.2$ | $\sigma = 1.8$ |
| ResNet-18 | ZeroPur-V-C-F | **69.52** | 55.77 | 55.77 | 62.36 | 46.09 | 34.39 |
| | ZeroPur-B-C-F | **74.12** | 65.59 | 65.59 | 65.43 | 69.56 | 60.14 |
| | ZeroPur-S-C-F | 69.58 | 51.66 | 51.66 | 77.56 | **85.15** | 78.93 |
| WRN-28-10 | ZeroPur-V-C-F | **68.82** | 47.94 | 47.94 | 59.19 | 36.27 | 29.80 |
| | ZeroPur-B-C-F | 66.39 | 60.59 | 60.59 | 58.88 | **70.66** | 58.75 |
| | ZeroPur-S-C-F | 62.75 | 41.79 | 41.79 | 73.68 | **83.92** | 77.24 |

Table 9: Impact of blurring operators in our method on robust accuracy (%) against PGD-20 $\ell_\infty$ threat model ($\epsilon = 8/255$), where we evaluate on ResNet-18 and WideResNet-28-10. Each row with the highest robust performance is bolded to indicate the most appropriate blurring operator.

## A.4 LIMITATION

One major limitation of our method is that method decreases the accuracy of the classifier when applied to clean natural images. We sample different subsets of ImageNet to verify this phenomenon and discuss this in Table 10. We use green to indicate the decreased natural accuracy with purification and red to indicate the gap between robust and natural accuracy with purification. We can see that for clean natural examples, the blurring operators corrupt their embeddings, so coarse shifting slightly decreases the accuracy of examples. Then, for fine alignment, which requires examples of orientation guidance, the coarse shifting results are no longer informative due to the blurring effect, and the accuracy of fine alignment will be further reduced.

| Dataset | Coarse Shifting | | Fine Alignment | | No Defense | |
|---|---|---|---|---|---|---|
| | Natural | Robust | Natural | Robust | Natural | Robust |
| Subset (correct) | 72.36 (-27.64) | 75.12 (+24.88) | 72.16 (-27.84) | 79.18 (+20.82) | 100.00 | 0.00 |
| Subset (random) | 62.15 (-18.25) | 64.02 (+16.38) | 47.33 (-33.07) | 67.54 (+12.86) | 80.40 | 0.00 |
| All | 61.96 (-18.42) | 63.96 (+16.42) | 46.93 (-33.45) | 67.36 (+13.02) | 80.38 | 0.00 |

Table 10: Natural accuracy (%) and Robust accuracy (%) against AutoAttack $\ell_\infty$ threat model ($\epsilon = 8/255$) on ImageNet or its subsets, obtained by the ResNet-50 classifier architecture. 'correct': 10k images that were correctly classified. 'random': 10k randomly selected images. 'all': ImageNet validation set. 'green': Decreased natural accuracy with purification. 'red': The gap between robust and natural accuracy with purification.

In this special case of $\mathbf{x}_{adv} = \mathbf{x}$, Equation 8 can be written as $\Delta \mathbf{u}'_l = f_l(\mathbf{x}') - f_l(\mathbf{x})$, and then we maximize $\mathrm{MILD}(\mathbf{x}_{adv}, \mathbf{x}', \mathbf{x}'')$. Since $\mathbf{x}'$ is a blurred counterpart of the natural example $\mathbf{x}$, the direction of $\Delta \mathbf{u}'_l$ is inherently wrong. However, fine alignment obtains the final purification result $\mathbf{x}''$ by moving the current result $\mathbf{x}'$ along $\Delta \mathbf{u}'_l$, which further leads incorrect movement. A promising solution is to borrow Test-time adaptation (TTA) Wang et al. (2020); Niu et al. (2022) techniques which seek to tackle potential distribution shifts between natural examples and adversarial examples by adapting our method, and we leave it for our future work.

## A.5 MORE RESULTS FOR ZEROPUR

**Rationalization of iterative blurring** In coarse shifting, we suggest that the blurring of a single time for adversarial examples should be attenuated, and the distance should be iteratively closed. Table 11 justifies this conclusion. We investigate the impact of gradually varying the strength of Gaussian blur ($\sigma$) and steps of coarse purification for the robust accuracy (%) on CIFAR-10, where the base classifier is ResNet-18. The '*' in 'Coarse-' denotes the iteration steps in coarse shifting.

We observe that (i) If we just filter examples (first column), the larger $\sigma$ would be better. However, when $\sigma$ is large, the samples are usually too vague to be recognized by the base classifier. (ii) If the filtering is done iteratively and then the distance is pulled, a small $\sigma$ is preferred as well as a moderate step. Consequently, in coarse shifting, slightly blurring samples are not enough to destroy the adversarial perturbations, but severely blurring samples tend to affect the decision of the base classifier. We suggest pulling the distance between adversarial examples and their blurred counterparts iteratively.

Table 11: The impact of gradually varying the Gaussian blur and steps of coarse purification for the robust accuracy (%) on CIFAR-10, where the base classifier is ResNet-18. The '*' in 'Coarse-' denotes the iteration steps in coarse shifting.

| $\sigma$ | Blur | Coarse-1 | Coarse-5 | Coarse-10 | Coarse-50 | Coarse-10 w/ Fine |
|---|---|---|---|---|---|---|
| 1.2 | 27.09 | 0.00 | 38.01 | 56.21 | 48.06 | **69.56** |
| 1.6 | 39.82 | 0.00 | 30.55 | 50.57 | 43.66 | **63.17** |
| 1.8 | 39.14 | 0.00 | 27.19 | 47.18 | 40.47 | **60.14** |
| 2 | 36.68 | 0.00 | 24.11 | 43.68 | 37.22 | **56.53** |

**Impact of purification steps** We continue to explore how the number of iterations affects the coarse shifting and fine alignment for the purification results, as illustrated in Figure 6. We can see that the number of iterations of the coarse shifting has a greater impact on purification results since the embeddings of blurred images are not exactly correct. On the contrary, the number of iterations of fine alignment has less influence on purification results, and it is mainly guided by the coarse shift. Thus in practice, we recommend setting a small number of iterations for coarse shifting.

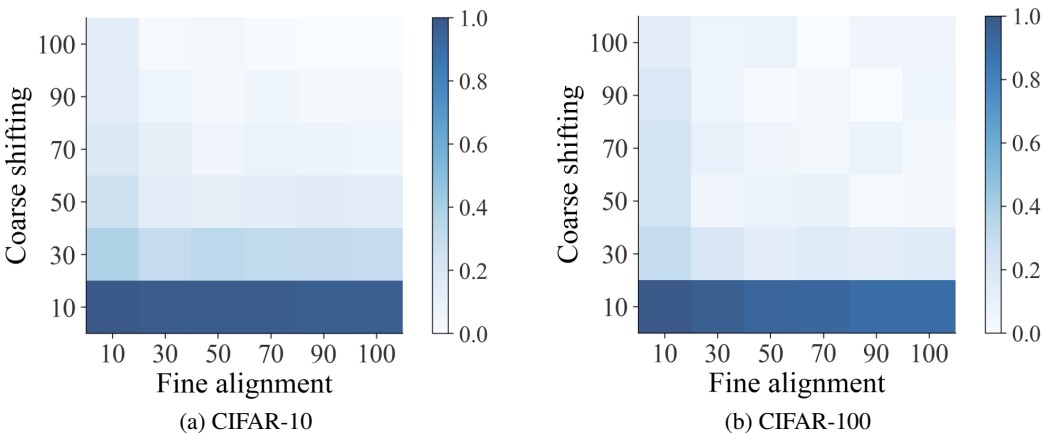

(a) CIFAR-10            (b) CIFAR-100

Figure 6: Impact of the number of iterations for coarse shifting and fine alignment. All robustness accuracy (%) are scaled to $[0, 1]$, with darker colors indicating better accuracy.

**Multiple Intermediate-Level Discrepancies in coarse shifting** We can naturally apply MILD to Equation 7:

$$d^* = \frac{1}{\|S\|} \sum_{l \in S} d([\mathbf{z}_{\mathrm{adv}}]^l, [\mathbf{z}'_{\mathrm{adv}}]^l) , \tag{12}$$

where $[\cdot]^l$ is the feature map in $l$-th layer.

Since 'Stage 5' is the global average pooling layer that aggregates all channels, we consider the sets that contain or do not contain it separately (e.g., Stage 3-4 and Stage 3-5). Compared to the baseline in the paper (coarse shifting with the single-layer features), multiple-layer features do slightly increase the upper bound for coarse shifting and fine alignment. The huge computational cost of frequently using multiple stages, however, is not a trade-off for a reasonably robust accuracy boost,

Table 12: Robust accuracy (%) against PGD-20 in $\ell_\infty$ threat model on CIFAR-10, where the architecture of base classification is ResNet-18. The highest accuracy is bolded.

| Method | Paper | Stage 3-4 | Stage 2-4 | Stage 1-4 | Stage 3-5 | Stage 2-5 | Stage 1-5 |
|---|---|---|---|---|---|---|---|
| ZeroPur-V-C | 53.73 | **57.87** | 54.81 | 51.17 | 52.21 | 52.35 | 52.30 |
| ZeroPur-V-C-F | 69.52 | 74.80 | 75.20 | **75.42** | 71.07 | 70.81 | 70.89 |
| ZeroPur-B-C | 56.21 | **61.25** | 60.52 | 59.09 | 57.02 | 57.02 | 57.00 |
| ZeroPur-B-C-F | 69.56 | 69.67 | 70.08 | 69.85 | 70.58 | 70.37 | **70.67** |
| ZeroPur-S-C | 77.64 | 79.14 | **79.85** | 79.55 | 77.93 | 78.11 | 77.74 |
| ZeroPur-S-C-F | **85.15** | 80.45 | 80.50 | 80.32 | 82.87 | 82.56 | 82.34 |

which defeats the keystones of being lightweight. Meanwhile, there is no uniform standard that can determine which layers are beneficial for each base classification. To summarize, we do not use it.

## A.6 VISUALIZATIONS

We show some results of ZerPur on ImageNet, obtained by the ResNet-50 classifier architecture. As illustrated in Figure 7 and Figure 8, the middle columns are the results of coarse shifting and fine alignment, while the red label is the error prediction and the green label is the correct prediction. As shown in Figure 9, we also show the case of failed purification. In the first row, the coarse shifting would have given the correct result, but the fine alignment instead pulled the example in the wrong direction. In the second row, although the fine alignment changes the prediction label, it is still not enough to push the example back within the natural image manifold.

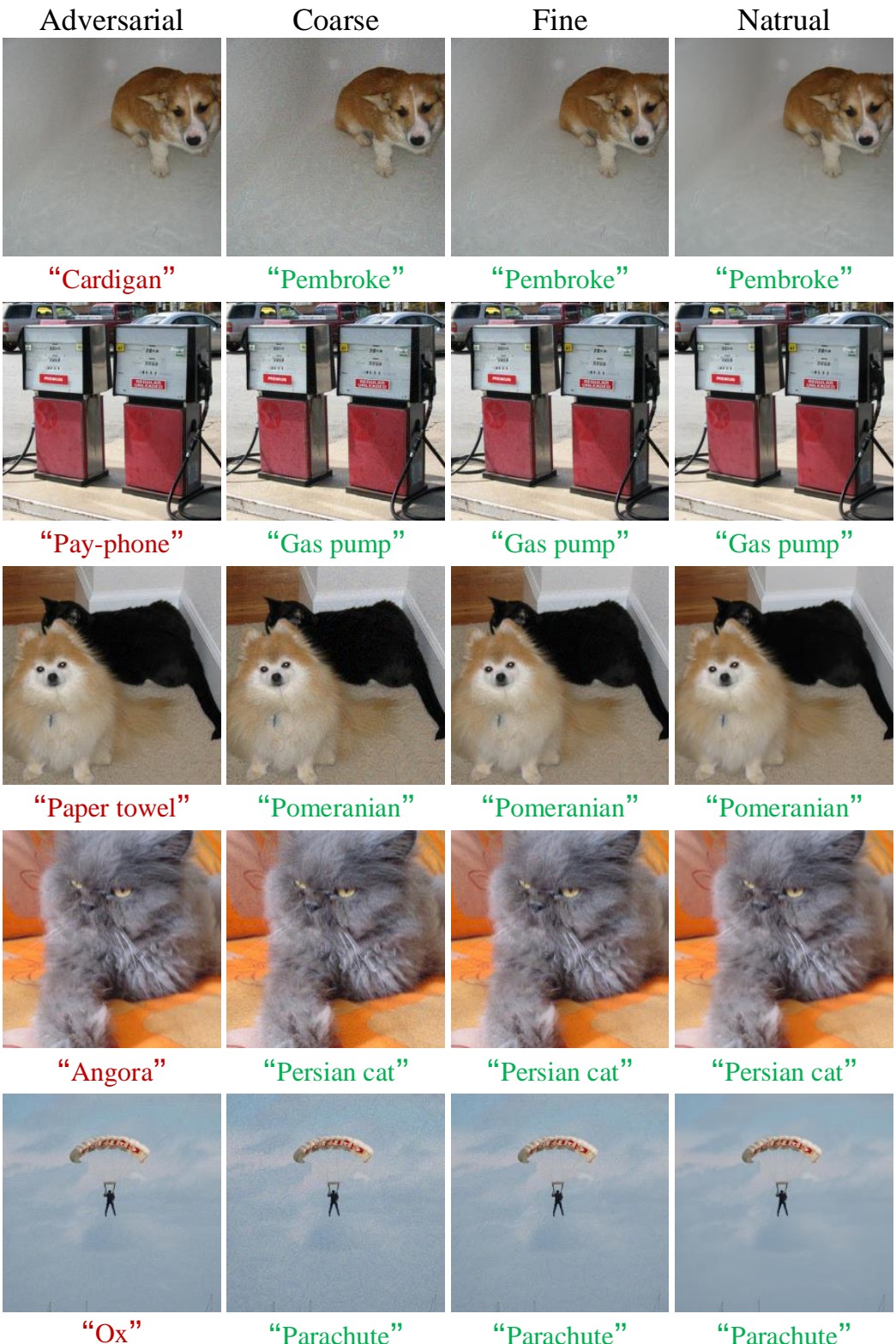

Figure 7: Visual example of ZeroPur against $\ell_\infty$ threat model ($\epsilon = 8/255$) on ImageNet. These adversarial examples are successfully purified directly by coarse shifting.

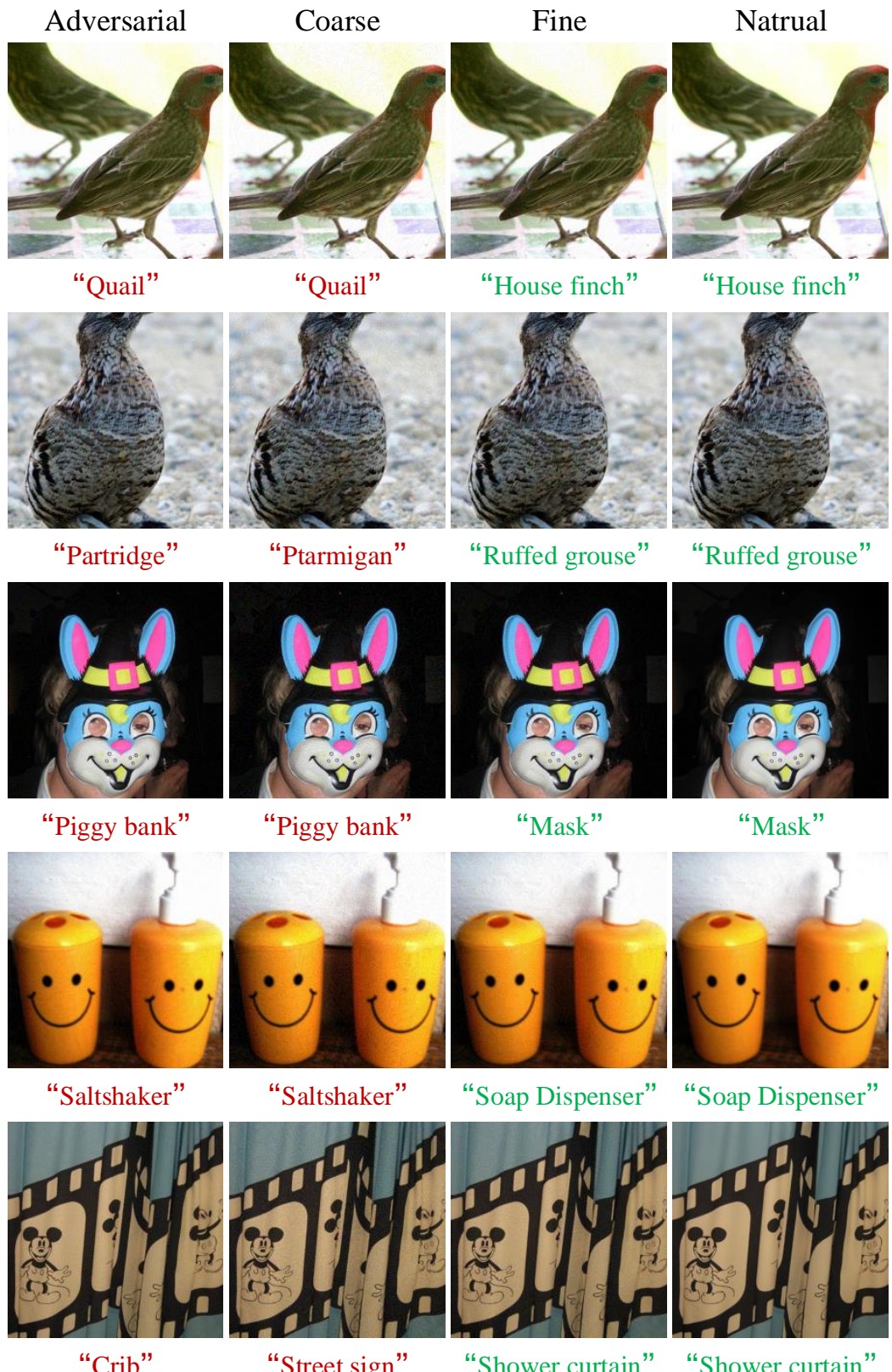

Figure 8: Visual example of ZeroPur against $\ell_\infty$ threat model ($\epsilon = 8/255$) on ImageNet. These adversarial examples need to be purified in two stages including coarse shifting and fine alignment.

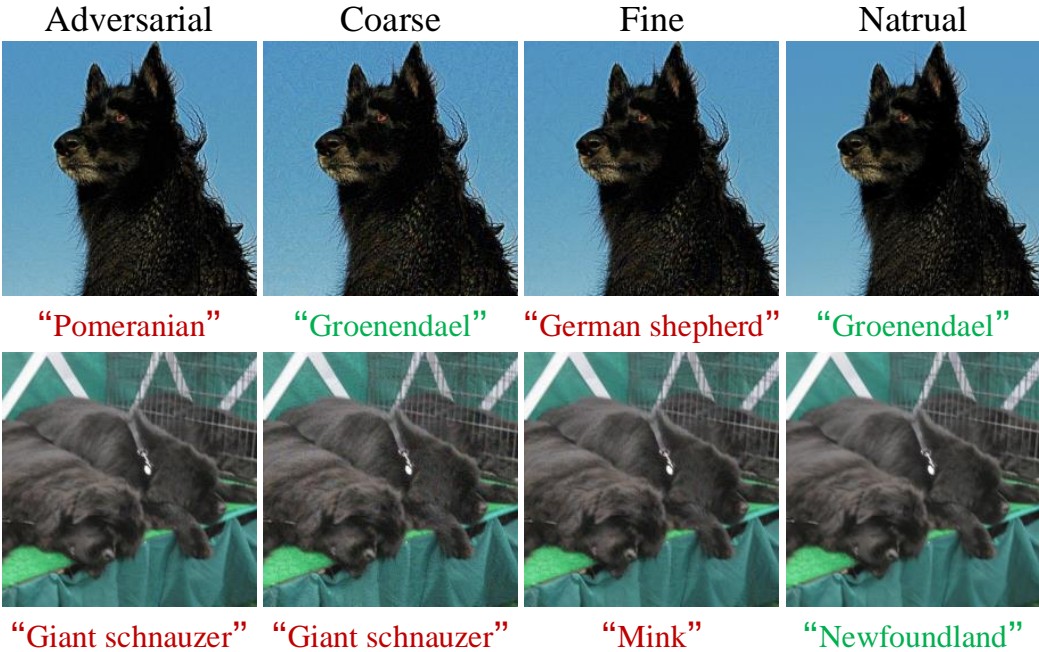

Figure 9: Visual examples of failures of purification.

