# OpenReview forum: "Towards a Self-Made Model: Zero-Shot Self-Supervised Purification for Adversarial Attacks"
_ICLR.cc/2024/Conference — Submitted to ICLR 2024_

### Official Review · Reviewer_GavC · 2023-10-30

**Soundness:** 2 fair
**Presentation:** 2 fair
**Contribution:** 2 fair
**Rating:** 5
**Confidence:** 4

**Summary:**

This paper focuses on zero-shot purification for adversarial attacks. Different from previous purification methods which require careful training of a strong generative model or incorporating additional knowledge when training a classifier to be comparable to adversarial training, this paper proposes a zero-shot self-supervised method for adversarial purification named ZeroPur, to purify the adversarial examples from coarse to fine. Following the natural image manifold hypothesis, the corrupted embedding vector of adversarial examples are first restored to be close to the natural image and then fine-tuned to project back within the manifold. Extensive experiments on different datasets including the ImageNet are conducted to demonstrate the effectiveness of the proposed method.

**Strengths:**

1. This paper focuses on adversarial purification which is more practical and low-cost than adversarial training for defending against adversarial attacks.
2. This paper proposes a new method, namely ZeroPur, that utilizes a blurred counterpart of the original adversarial images to purify the adversarial input, which has some technical contributions for enhancing the plained adversarial purification method.
3. This paper present comprehensive experimental results on different benchmark datasets to demonstrate the effectiveness of the proposed ZeroPur.

**Weaknesses:**

Overall, this work presents a new purification method for effectively defending against adversarial examples. Here are the major concerns for the current version of this paper, and hope it can help to improve the paper better and address the potential concerns.
1. In the motivation parts, the proposed method is mainly based on the empirical investigation of different augmentation methods for adversarial examples. As the authors also stated in section 3.2, the motivation is a little bit heuristic. It is concerned whether it is general enough to different types of unseen attacks, since there is limited general guidance or explanation about the underlying mechanism of the observation. Especially why the adversarial examples that deviated from the natural image manifold are coming back closer to the manifold.
2. The critical part of the proposed method is not intuitive and not easy to understand why it would be effective. As previous concern stated, the method is based on the heuristic investigation, the current presentation does not reveal the underlying mechanism of such a defense. Without any theoretical justifications, the reliability of this kind of method is questionable. The current version of the presentation makes readers hard to get an intuitive idea about whether it is general enough for defending against adversarial examples, or it only works on the image domain with the blurring technique.
3. Technically, the proposed method contains the coarse shifting and fine alignment part, for the fine alignment part, the algorithm seems to also train the models, which is not consistent with the zero-shot claim. Could the author provide more explanation on this part to address the potential misunderstanding? In addition, why this work needs to emphasize the zero-shot concepts?
4. For the experimental part, could the author provide a more detailed procedure about how to adopt the AutoAttack in the newly proposed method, it could be better to clearly plot the evaluation procedure like Diffpure shown in Figure 1 to make the evaluation more convincing, since the pre-processing based defense may need carefully reviewing, especially for its evaluation part on adaptive attacks.
5. The proposed method also contains additional knowledge on the design of coarse shifting and fine alignment part since it utilize the augmentation example like the blurring image, please provide a more detailed definition of additional knowledge. Otherwise, some conclusions or statements of the current version is overclaimed.

**Questions:**

The following questions are more specific to the clarification and the reviewer hope these question or comments may help the authors to improve the writing and presentation of this work.
1. In the abstract, "such methods can ... ... in contrast to empirical defenses". Without any theoretical guarantees or justification, adversarial purification is also a kind of empirical defense method, instead of a certified or theoretical-supported one. It may be not appropriate to state with "in contrast to the empirical defense". Please also revise this sentence.
2. The "natural image manifold hypothesis" is not clearly defined or referred to in the introduction and the following sections. It could be better to provide a rigorous definition or some citations to clearly present the related parts.
3. The technical contribution does not closely correspond to the previous challenges stated in the third paragraph of the introduction. It is hard to catch the point why the newly proposed method can achieve satisfactory performance and also not rely on training promising generative models.
4. Using the simple blurring operator seems to belong to the pre-processing methods for adversarial defense, however, the current related work part does not contain the discussion about related literature on this research direction.
5. It could be better to explain the underlying mechanism of using such data augmentation methods to encourage the adversarial examples coming back to the natural manifold. The corresponding presentation in section 3 can be further enhanced.
6. please clearly provide its evaluation on the adaptive attacks.

---

### Official Review · Reviewer_tryU · 2023-10-30

**Soundness:** 2 fair
**Presentation:** 3 good
**Contribution:** 2 fair
**Rating:** 3
**Confidence:** 4

**Summary:**

The paper presents an adversarial purification method that relies on using a blurred version of the input adversarial image as a target for purification. The proposed algorithm is divided into two steps: 1. A coarse purification step that pushes the input image embeddings closer to those of a blurred copy, and 2. further finetuning the embeddings to match those of the difference between the coarse purified image and the adversarial image, The algorithm is evaluated with a variety of white-box and BPDA style attacks.

**Strengths:**

1. The algorithm presents better performance when evaluated with attacks like BPDA+EOT, as well as whitebox attacks.
2. The coarse purification part is well-motivated and evaluated.
3. The paper clearly mentions the limitations; specifically with regards to natural accuracy being affected by the approach

**Weaknesses:**

1 The writing is unclear in some sections; for example "The increase of projection implies that x
′′ is not restricted by the blurring example to continue moving along the direction of the coarse result,which makes x
hopefully move independently to the natural manifold.".

2. The finetuning objective maximizes the similarity between the coarse purified image while minimizing the same with the adversarial image. However, a similar effect can be possibly achieved with learning rate scheduling or a learning rate search method. Can the authors comment why the additional finetuning step is better?

3. Following Croce et al, Evaluating the Adversarial Robustness of Adaptive Test-time Defenses, ICML 2022, I also suggest the authors test the defense with transfer attacks from the base model as well as black box attacks to rule out gradient obfuscation.

4. "We believe that the direct use of strong adaptive attacks underestimates the robustness of ZeroPur" - can the authors clarify this statement? The point of having strong adaptive attacks is to clearly show the efficacy of the proposed defense. In this case, if an adaptive attack breaks the defense, the robustness is clearly not underestimated. If the authors mean to suggest a different threat model, it would be great to have clear definitions.

**Questions:**

Some additional questions I have;
1. What is the compute overhead when compared to other purification methods like Aid-purifier (Hwang et al, 2021) and Anti-advesaries (Alfarra 2022)?

---

### Official Review · Reviewer_1obD · 2023-11-04

**Soundness:** 1 poor
**Presentation:** 1 poor
**Contribution:** 2 fair
**Rating:** 3
**Confidence:** 2

**Summary:**

The paper presents a purification method to defend adversarial attack. The main idea is to iteratively change the input (supposed to be an advesarial example) such that the embedding of the input (with a classifier) is close to the embedding of the corrupted input. The corruption could be performed with any fixed methods. The idea comes from the observation that a simple blurring or some data augmentation methods could defend adversarial attack to some extent.

**Strengths:**

The idea is simple. The motivation from the observation to the proposed method is clear. The robust accuracy is high according to the reported experimental results.

**Weaknesses:**

The proposed method does not make sense because it doesn't consider the accuracy of clean examples, which is very important for a deep learning model. In most experiments, this clean accuracy is not reported, except on ImageNet (not good on this dataset). It is stated in the conclusion that this accuracy is not high. This is expected because image corruption definitely degrades the recognition accuracy.

The proposed method uses data augmentation. It is well known that data augmentation can improve adversarial robustness (e.g., [a]). It is unknown how much improvement reported in the paper comes from data augmentation.
[a] Lin Li and Michael W. Spratling. Data augmentation alone can improve adversarial training. ICLR 2023.

The proposed method requires to iteratively adjust the input according to the output of the classifier, which definitely increases the inference speed. So the speed is in general K times slower than other defense methods such as adversarial training methods, where K denotes the number of iterations.

The presentation is unclear. It is hard to understand the method. Below I list some unclear points:
1. The loss functions are widely used in the paper but not defined. In eqn (1) the loss function has two arguments but in eqn (6) another loss function has one argument. The definitions of the loss functions are important in a paper about adversarial defense like this one, because sometimes the loss functions are minimized and sometimes are maximized. Without a clear definition, it is hard to understand the whole paper.
2. Figure 2 presents the "responses" of different classifiers, but this term is undefined. How do you get the response on an input image? I guess the authors use certain interpretation method to do the visualization, but this should be stated clearly.
3. Figure 2 has (a), (b) and (c), and each subfigure has different columns. In the caption, there are words: left, middle and right. Do they correspond to the three subfigures or the columns inside different subfigures?
4. In Figure 3, what are A, A' and N?
5. In page 4, the beginning of the paragraph that has eqn (6): "However, the natural example x and its label y are invisible..." Do you mean during the inference phase? During training, I think this information is known. The method governed by eqn (6) is about training or inference? Such basic but important message should be provided explicitly.
6. Since the important notations u' and u'' in sec. 3.3 are undefined, I cannot understand this part of the method.
7. The paper reports results of the proposed method with "retraining" in several tables. But it is not described how this retraining is performed.

**Questions:**

I may raise the score if clean accuracy and inference speed are comparable with SOTA methods.

---

### Meta-Review · Area_Chair_MDjv · 2023-12-04

**Metareview:**

The reviewers show negative attitudes toward the current submission. And the authors do not reply.

**Justification For Why Not Higher Score:**

N/A

**Justification For Why Not Lower Score:**

N/A

---

### Decision · Program_Chairs · 2024-01-16

Reject